# Global and regional ocean mass budget closure since 2003

Carsten Bjerre Ludwigsen [1,2] ✉, Ole Baltazar Andersen [1], Ben Marzeion [3], Jan-Hendrik Malles[3], Hannes Müller Schmied [4,5], Petra Döll [4,5], Christopher Watson[2,6] & Matt A. King [2,6]

In recent sea level studies, discrepancies have arisen in ocean mass observations obtained from the Gravity Recovery and Climate Experiment and its successor, GRACE Follow-On, with GRACE estimates consistently appearing lower than density-corrected ocean volume observations since 2015. These disparities have raised concerns about potential systematic biases in sea-level observations, with significant implications for our understanding of this essential climate variable. Here, we reconstruct the global and regional ocean mass change through models of ice and water mass changes on land and find that it closely aligns with both GRACE and density-corrected ocean volume observations after implementing recent adjustments to the wet troposphere correction and halosteric sea level. While natural variability in terrestrial water storage is important on interannual timescales, we find that the net increase in ocean mass over 20 years can be almost entirely attributed to ice wastage and human management of water resources.

Accurately quantifying the contributions to global and regional sea level change is essential for understanding one of the great societal threats resulting from climate change, with hundreds of millions worldwide vulnerable to changing coastal sea level[1]. The primary reason for sea level changes, both globally and within most ocean regions, is the increase in the exchange of water mass from land into the ocean[2–4], something that is expected to increase in the future[5–9]. Accurately quantifying the contributions to global and regional ocean-mass change is, therefore, essential for understanding and predicting future regional sea levels.

The mass change of both land and ocean is inferred from near-continuous time-varying estimates of Earth's gravity field by the Gravity Recovery and Climate Experiment (GRACE, 2003–2017) and its successor, GRACE Follow-on (GFO, 2018-) satellite missions. The GRACE-GFO-derived time series of ocean mass (Supplementary Fig. 1) shows a steady increase in mass since 2003, consistent with increased sea levels[4,10,11], and ice sheet[12] and glacier[13,14] mass loss. Since 2016, the mass change observed by GRACE has, however, significantly flattened compared to pre-2016 (Supplementary Fig. 1) period, which coincides with increased GRACE battery maintenance in 2016[15] followed by a 13-month gap, before the launch of GFO. The accuracy of the observed flattening in GRACE-GFO needs to be confirmed by independent techniques.

We can calculate ocean mass changes using altimetric sea surface height observations, but only after subtracting the steric sea level changes derived from ocean temperature and salinity data[16–19]. We refer to this approach as 'steric-corrected altimetry'. Alternatively, we can reconstruct ocean mass changes by summing the contributions from land, including changes in water storage on continents as liquid water and snow (land water storage, LWS) and changes in land ice[4,20], which can be derived from ice and land surface models or alternative observing techniques, such as ice-sheet altimetry. Our reconstruction is hereafter referred to as "OMrecon". The process of comparing two or three independent estimates of ocean mass change is referred to as "closing the ocean-mass budget"[17].

[1]Technical University of Denmark, DTU Space, Lyngby, Denmark. [2]The Australian Centre for Excellence in Antarctic Science, University of Tasmania, Hobart, TAS 7001, Australia. [3]Institute of Geography and MARUM—Center for Marine Environmental Research, University of Bremen, Bremen, Germany. [4]Institute of Physical Geography, Goethe University Frankfurt, Frankfurt am Main, Germany. [5]Senckenberg Leibniz Biodiversity and Climate Research Centre (SBiK-F), Frankfurt am Main, Germany. [6]School of Geography, Planning, and Spatial Sciences, University of Tasmania, Hobart, TAS 7001, Australia. ✉ e-mail: caanlu@space.dtu.dk

In the last few years, several studies have attempted to close the ocean mass or sea-level budget on both global[2–4,11,21–25] and regional[2,3,17,26] scales by mainly comparing GRACE and steric-corrected altimetry. A few of these studies have combined ice and hydrological models to estimate mass-driven sea level changes and compared these with observations from GRACE and altimetry[4,21,24,25]. Commonly, these studies show that the global sea level or ocean-mass budget closes from 2005 to 2015, when GRACE was operating nominally[27] and the autonomous Argo observation system was operating at full capacity[28], while only partially balancing the budget at regional scales[2,17]. However, more recent studies[18,22,23,29,30] have not been able to confirm the flattening observed by GRACE-GFO after 2016, suggesting that GRACE-GFO underreports the ocean mass change following instrumental issues after 2016[15]. Concerningly, the ocean mass budget did not return to closure[22,23] after GFO became operational in mid-2018, thereby leading to a lack of closure of the sea level/ocean mass budget since 2016. This raises questions about each observational technique and our understanding of the global water cycle[31].

In this study, we develop an improved reconstruction, based on modeled and to a large extent independent data and consider recently detected systematic errors, to provide three largely-independent ocean mass datasets with monthly temporal resolution and global coverage over 2003–2022 inclusive, thereby encompassing 20 years of observations.

We first compare the GRACE-GFO mass time series to entirely independent steric-corrected altimetry data. We apply the Wet Troposphere Correction (WTC) obtained from the satellite onboard microwave radiometer (MWR)[32] to sea surface height observations from altimetry and remove the contribution from Glacial Isostatic Adjustment[33] and account for ocean bottom deformation caused by current land mass changes[31,34]. The steric sea level change signal has been computed down to 5400 meters accounting for temperature and salinity changes over the near full-depth of the ocean column. We note that the deep ocean below 2000 m is showing near constant warming, which has a non-negligible contribution to steric sea level change[35] (Supplementary Fig. 2). The steric sea level change has been adjusted to account for a well-known issue, the erroneous salinity drifts in the ARGO observing system[22,36,37], by removing the annual global-mean halosteric change while retaining the seasonal halosteric signal (see Methods for details). The mass time series from steric-corrected altimetry with the default MWR WTC shows good agreement with GRACE from 2005 to 2016 but shows increased discrepancy after 2016 (dotted purple line in Fig. 1c), with a post-2016 rate difference of

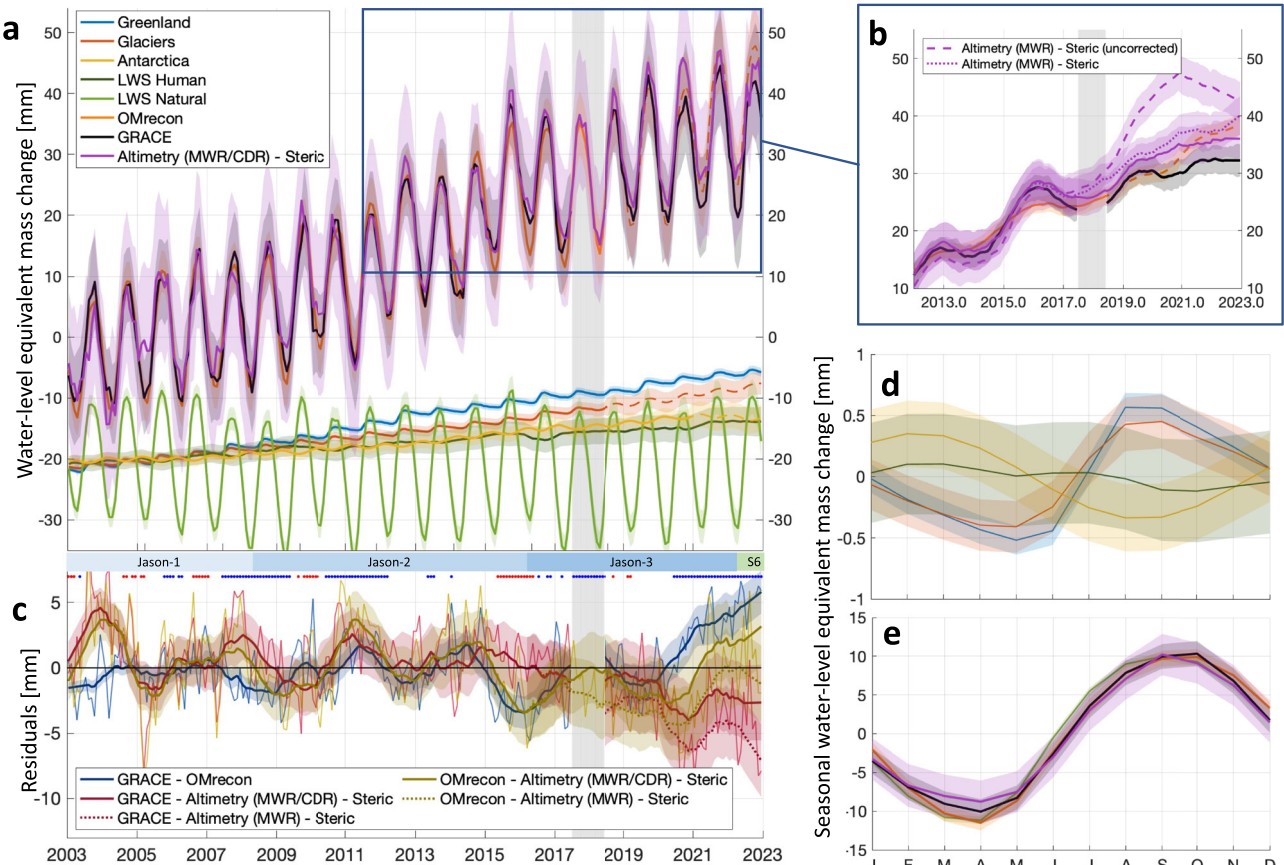

**Fig. 1 | Ocean mass budget timeseries from 2003 to 2022. a** Monthly global ocean mass anomalies from 2003 to 2022, referenced to the 2003–2008 mean for the three ocean mass estimates and five land contributions to OMrecon (offset by −20 mm for clarity). Dashed lines indicate periods where the mass change has been extended from the original data. The colored bars below the panel indicate the periods for the different altimetry missions (S6 = Sentinel-6 Michael Freilich). **b** Same as a, but 12-month averaged and with the addition of steric-corrected altimetry with the default Microwave Radiometer (MWR) Wet Troposphere Correction (WTC) applied, without (dashed, light purple line) and with (dotted, dark purple line) the halosteric drift correction. **c** Monthly (thin lines) and 12-month averaged residuals (thick lines) between Omrecon, GRACE, and steric-corrected Altimetry with and without MWR correction (dotted lines). Blue and red dots indicate months affected by La Niña and El Niño, respectively. Gray areas in (**a**–**c**) indicate periods with no GRACE observations. **d** Averaged seasonal mass anomaly for the three ice mass contributions and effect of human water management on naturalized land water storage. **e** Same as (**d**), but for naturalized land water storage and the three ocean mass estimates on a different scale. **a**–**e** All values are provided in mm global ocean mass change and shaded areas indicate 1σ uncertainties. Data for this figure is provided in the Source Data file.

**Table 1 | Components of the sea level budget**

| Global ocean mass budget from 01/2003 to 12/2022 | Trend ± 1σ [mm y⁻¹] | Phase ± 1σ [deg] | Amplitude ± 1σ [mm] |
|---|---|---|---|
| Greenland (incl. peripheral glaciers) | 0.85 ± 0.04 | 276.5 ± 9.3 | 0.48 ± 0.04 |
| Antarctica (incl. peripheral glaciers) | 0.43 ± 0.07 | 56.3 ± 11.2 | 0.35 ± 0.03 |
| Glaciers (excl. pheriphery of ice sheets) | 0.67 ± 0.07 | 269.8 ± 7.1 | 0.40 ± 0.02 |
| Land water storage (human) | 0.35 ± 0.05 | 92.5 ± 36.5 | 0.09 ± 0.04 |
| Land water storage (natural) | 0.01 ± 0.14 | 260.2 ± 1.7 | 10.84 ± 0.19 |
| Sum of contributions (Barystatic) | 2.32 ± 0.18 | 260 ± 1.0 | 11.64 ± 0.31 |
| Global mean atmospheric mass | −0.09 ± 0.02 | 195.0 ± 2.0 | 1.55 ± 0.07 |
| OMrecon (Barystatic + Atmosphere) | 2.23 ± 0.19 | 269.4 ± 1.8 | 10.78 ± 0.20 |
| GRACE | 2.11 ± 0.14 | 266.4 ± 2.8 | 10.33 ± 0.26 |
| Steric | 1.49 ± 0.22 | 62.7 ± 2.5 | 4.67 ± 0.67 |
| Glacial Isostatic Adjustment (GIA) | −0.24 ± 0.06 | N/A | N/A |
| Ocean Bottom Deformation (OBD) | −0.18 ± 0.02 | 70.6 ± 1.0 | 0.99 ± 0.02 |
| Altimetry (MWR / CDR) - Steric - GIA - OBD | 2.28 ± 0.36 | 266.4 ± 3.4 | 9.52 ± 0.31 |
| Altimetry (MWR-only) - Steric - GIA - OBD | 2.39 ± 0.34 | 266.9 ± 3.4 | 9.50 ± 0.30 |
| OMrecon—GRACE | 0.12 ± 0.24 | 314.0 ± 77.0 | 0.76 ± 0.20 |
| OMrecon—Altimetry (MWR / CDR) | −0.05 ± 0.40 | 291.7 ± 22.1 | 1.39 ± 0.25 |

Linear trends, and annual phase and amplitude, of global ocean-mass change and its contributors over the period 01/2003 to 12/2022.
The trends, phase (phase zero is 00:00:00 on January 1st), amplitude, and associated uncertainties are computed from bootstrapping (see Methods).

0.91 ± 0.44 mm y⁻¹ (dotted pink line in Fig. 1b; all in text uncertainties are one standard deviation).

Recently, studies suggest[38] that the MWR of Jason-3 is drifting leading to WTC bias after Jason-3 enters the altimetry time series in March 2016. Following[25,38], we recalculate the WTC from water vapor measurements obtained from satellite-borne radiometers (distinct from altimetry satellites), which have been carefully intercalibrated and are suitable for long-term climate studies, called Climate Data Records (CDR)[38,39]. After applying the CDR-adjusted WTC for the Jason-3 period, the post-2016 rate difference with GRACE is reduced to 0.42 ± 0.45 mm y⁻¹ (similar to the results of ref. 25, Supplementary Fig. 3). Thereby, we reduce the post-2016 budget residual between steric-corrected altimetry and GRACE with agreement within one standard deviation until 2020, whereafter a significant divergent ocean mass change is still evident (Fig. 1). In our subsequent analysis (denoted with MWR/CDR), we adopt the steric-corrected altimetry with the annual global halosteric contribution removed and the modified WTC applied as the default steric-corrected altimetry.

## Results
### Reconstructing ocean mass
To further verify the ocean mass change observed by GRACE-GFO and steric-corrected altimetry, we compare with a third estimate, which reconstructs ocean mass changes from land hydrology and land ice (hereon termed OMrecon). OMrecon uses different time series of land-based water mass change separated into five sources; modeled glacier mass balances (excluding periphery)[14], modeled Greenland mass balance[40] and multi-method ensemble of Antarctica mass balance[12] including peripheral glaciers, and modeled natural and human impact on LWS[41]. Each of the five sources of land mass change is converted into spatial ocean mass change by considering Gravitational, Deformational, and Rotational (GRD) effects[42] on the ocean caused by loading change (see Supplementary Fig. 4 and Methods for details).

Mass-balance time series of glaciers and the Antarctic Ice Sheet terminates in December 2018 and 2020 respectively, and thus, a slightly shorter timespan is available compared to the GRACE observational record. To extend these mass contributions to the full duration of the GRACE and altimeter records, we extrapolate them using a seasonal decomposition model in conjunction with detrended gravitational measurements from GFO (see Methods for further detail). To account for land and ice mass changes that do not flux directly into the ocean, we subtracted global mean atmosphere mass change[43] from the sum of the five land sources to ocean-mass change.

### Global mass-driven sea level change
The reconstruction of ocean mass (Fig. 1 and Table 1) shows that between January 2003 and December 2022, the combined contributions of ice and land (OMrecon) have resulted in a change of 44.6 ± 3.8 mm in ocean mass water level equivalent. Among these contributions, Greenland (17.0 ± 0.8 mm) and Glaciers (13.4 ± 1.4 mm) are the primary contributors. When considering the loss of Antarctic ice (8.6 ± 1.4 mm) in conjunction with the ice change from Greenland and Glaciers, approximately 85% of the reconstructed ocean mass change is attributed to ice loss. The remaining ocean mass change is attributed to human water use and man-made reservoirs (LWS Human, 7.0 ± 1.0 mm), while natural land water changes contribute with a negligible ocean mass change over the 20 years (LWS Natural, 0.2 ± 2.8 mm).

These results agree with the GRACE observations that show a global ocean-mass change of 42.2 ± 2.8 mm water level equivalent. Steric-corrected altimetry observations without correcting for salinity drift yield a change of 55.8 ± 7.2 mm (Fig. 1b). This is reduced to 47.8 ± 7.2 mm when the annual global mean halosteric contribution is removed but is still a considerably higher mass change than observed by GRACE-GFO and OMrecon (Fig. 1b). Adopting a modified WTC calculated from climate data records (CDR)[38,39] for the period when Jason-3 is integrated into the altimetric record (March 2016 to May 2022) lowers the altimetry mass change estimates to 45.7 ± 7.3 mm over the 20 years. The global ocean mass change is thus in agreement within the range of one standard deviation, among all three estimates.

The residual monthly signals, as depicted in Fig. 1c, are predominantly within the range of ±5 mm, indicating a strong agreement until 2020. The introduction of the modified WTC significantly improved the agreement after the launch of Jason-3. This modification causes the steric-corrected altimetry data to shift from being outside the combined 1σ-uncertainty to being within it when compared with GRACE-GFO after the year 2016 (Fig. 1c).

OMrecon deviates from both GRACE and steric-corrected altimetry in the last 3 years of the time series with an average 1.75 ± 0.18 mm y⁻¹ trend difference from 2020 to 2022. This deviation

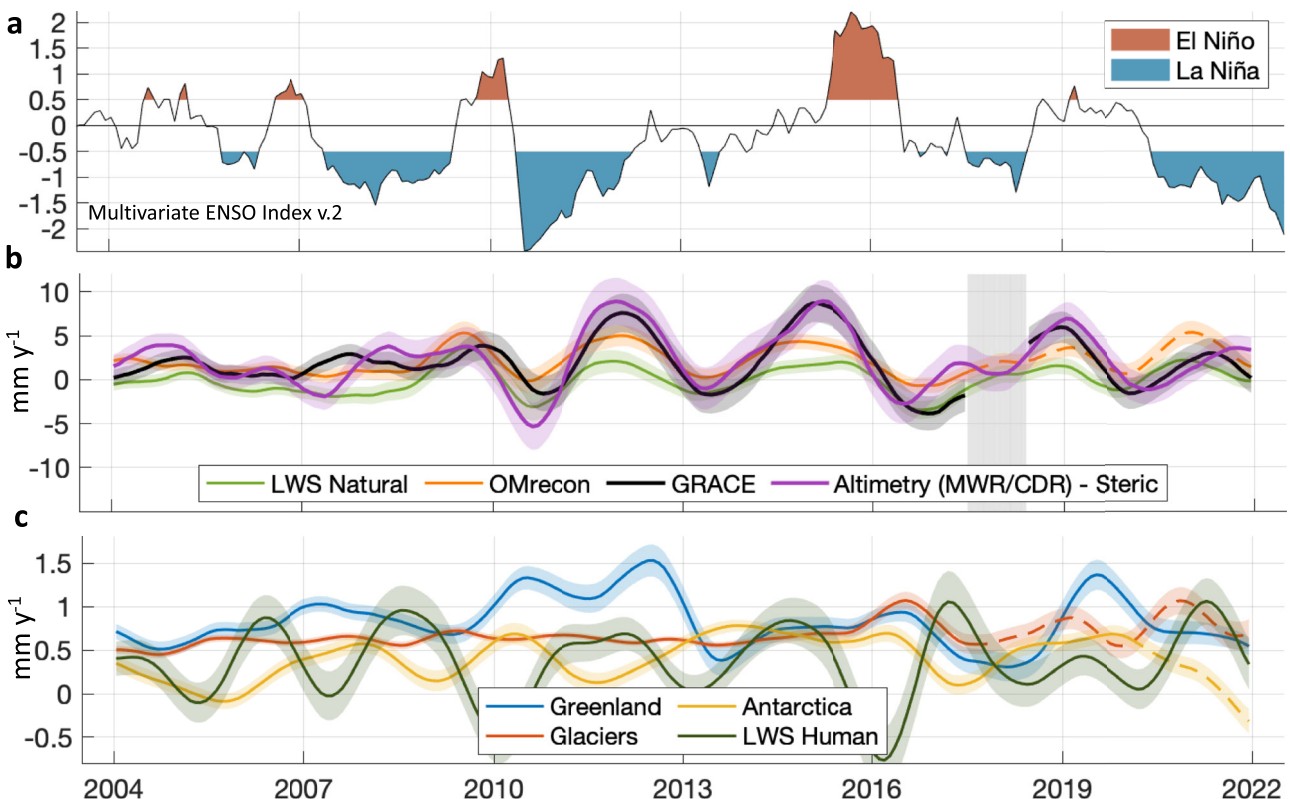

**Fig. 2 | Rates of ocean mass change. a** Multivariate El Niño–Southern Oscillation (ENSO) Index (MEI. v2). The ENSO phase is neutral for MEI. v2 index values between −0.5 and 0.5. **b** Ocean mass change rates (12-month average) for the three ocean mass estimates and natural land water storage (LWS). Dashed lines indicate periods, where the original products have been extended (see methods). **c** Same as (**b**), but for the three ice contributions and LWS human. Shaded areas indicate 1σ uncertainties. Data for this figure is provided in the Source Data file.

coincides with a strong negative phase (La Niña) of the El Niño-Southern Oscillation (ENSO), which generally enhances precipitation over land and consequently lowers ocean mass change[44]. The hydrological model[45] used to estimate LWS, exhibits a decreasing trend in land mass during the latest La Niña phase since 2020, mainly due to underestimating minimum mass in the northern hemisphere summers (Supplementary Fig. 5). However, the GFO-observed land mass changes, after correcting for ice mass, show a significant build-up of land mass (Supplementary Fig. 5). This contrasting pattern can be attributed to the tendencies of the climate reanalysis data[46], which drives the hydrological model to underestimate precipitation in mid-latitudes[47]. This is, in particular, observed on the African continent (see Supplementary Fig. 5), where the model underestimates the LWS change with ~1500 Gt (~4 mm water level equivalent) from 2019–2022. When we exclude the two mid-African regions (Supplementary Fig. 5), we found good agreement between modeled LWS change and GRACE observations over land. Consequently, we attribute the deviation between the OMrecon and the other two ocean mass estimates during 2020–2022 to the unaccounted La Niña effect in the contribution from natural land water storage.

We investigate the components of seasonal mass change, by fitting an annual sinusoidal function to each time series (Fig. 1d, e. and Table 1). The phase of the seasonal reconstruction from OMrecon agrees with both GRACE and steric-corrected altimetry within 1σ, while the amplitude of OMrecon driven by LWS agrees with GRACE-observations, but not fully with steric-corrected altimetry. It is evident that the seasonal signal is dominated by natural land water storage with only a minor (~10%) contribution from the cryosphere and human water management (Fig. 1d, e). The phase of OMrecon agrees within a few degrees with GRACE and steric-corrected Altimetry (as also shown by ref. 48), while OMrecon shows a larger seasonal amplitude (Table 1).

Over longer periods, the global ocean mass change has a strong inter-annual variation (Fig. 2), with maximum 12-month averaged rates reaching up to 10 mm y$^{-1}$ and a minimum rate of −5 mm y$^{-1}$. These inter-annual changes follow the ocean mass change rate caused by LWS change (Fig. 2b). Throughout the entire time series, it is apparent that OMrecon consistently underestimates the inter-annual variability of ocean mass change in comparison to GRACE and steric-corrected altimetry, as depicted in Figs. 1 and 2. This underestimation appears to be associated with El Niño and La Niña periods (Figs. 1c and 2) suggesting a general smoothing or underrepresentation of short-term climatic effects in the adopted hydrological models, which is in line with the findings of ref. 47.

The rate of ocean mass change was at its highest in 2012 and 2015, while it has slowed since 2016 (Fig. 2b). Since the end of 2016, the ENSO has mostly been in its negative phase, with a particularly strong and persistent La Niña since mid-2020 resulting in lower rates of ocean mass change since 2019. A small slowing of Greenland ice loss, after it reached a recent high in 2019–2020, correlates with a shift of the North Atlantic Oscillation[49] from being predominately negative to mostly positive at the end of 2019, which has also contributed to lower global ocean mass change rates since 2020.

## Regional ocean mass budgets

We next compare the three datasets over five ocean basins and a sixth region covering the remaining ocean area (Fig. 3 and Supplementary Figs. 2, 6–7). Overall, all regions have gained significant amounts of ocean mass since 2003, with trends ranging from 1.40 to 2.85 mm y$^{-1}$ across all regions and estimates. These trends are also all (except in one case) larger than the globally averaged steric sea level trend of 1.50 mm y$^{-1}$ (Table 1).

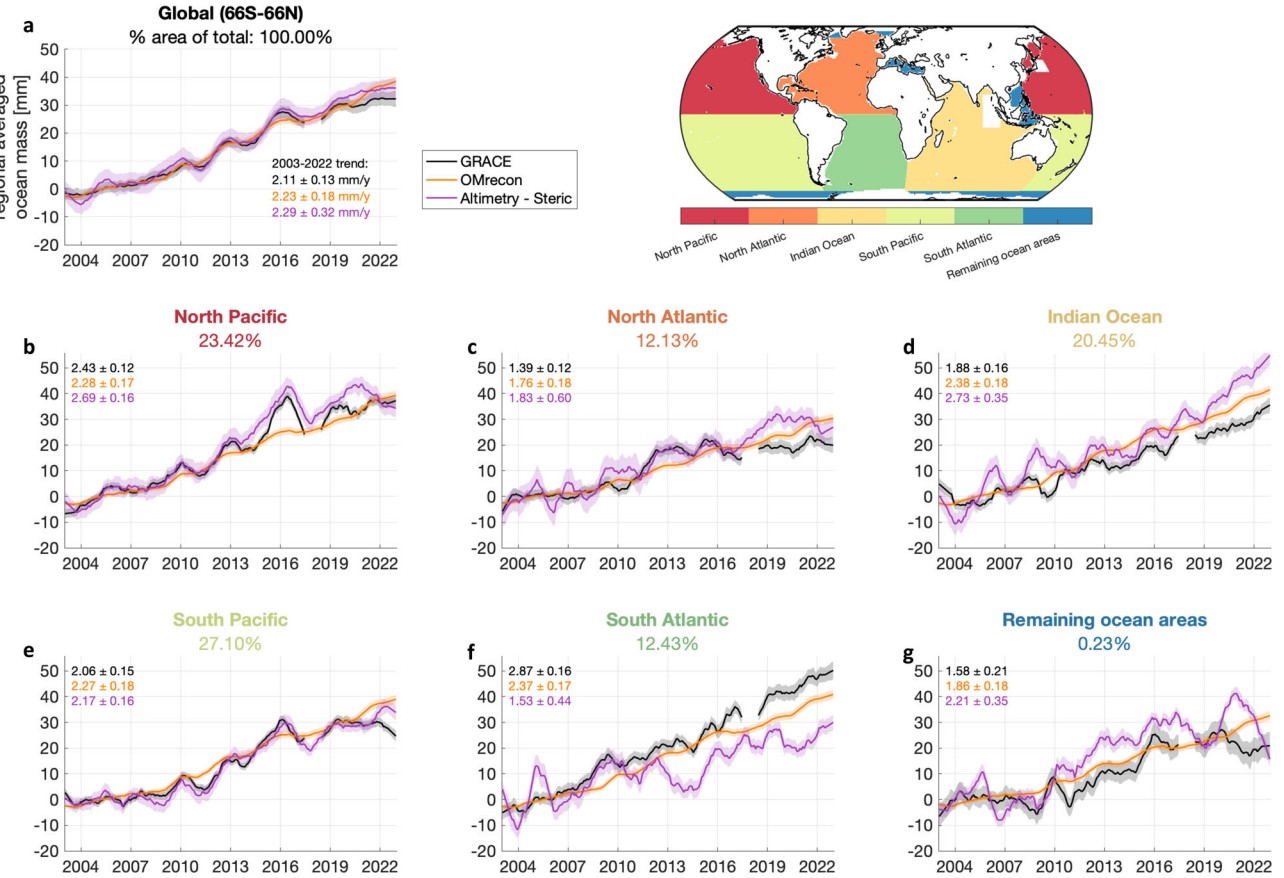

**Fig. 3 | Regional ocean mass budgets.** 12-month moving averages of the three ocean mass change contributions shown globally (**a**), for 5 ocean reg.-ions (**b**–**f**) and remaining ocean areas (**g**). The map indicates each ocean region and the global mask applied throughout this study. Shaded areas indicate 1σ uncertainties. The values are relative to the 2003–2008 mean. Data for this figure is provided in the Source Data file.

In general, OMrecon shows a consistent near-linear mass increase across all regions. On the other hand, the mass anomalies of GRACE-GFO and steric-corrected altimetry show greater interannual variation between regions. This variation is indicative of mainly wind-driven dynamic sea level changes as the main underlying cause.

However, it is worth noting that there are other contributors to inter-regional variations, which become evident when comparing GRACE-GFO and steric-corrected Altimetry. In particular, opposite differences are observed between the North and South Atlantic. Here, the GRACE trend is 1.45 mm y⁻¹ larger in the South Atlantic compared to the North Atlantic, while steric-corrected altimetry, in contrast, shows lower-than-average ocean mass change in the South Atlantic (Fig. 3 and Supplementary Fig. 6).

Combining the two Atlantic regions closes the Atlantic budget between GRACE (1.96–2.28 mm y⁻¹), OMrecon (1.88–2.26 mm y⁻¹) and steric-corrected Altimetry (1.31–2.06 mm y⁻¹) over the entire time-series. The difference between uncorrected and drift-corrected halosteric sea level change in the North Atlantic (Supplementary Fig. 2) agrees with the shown 2018–2022 discrepancy between GRACE-GFO and steric-corrected Altimetry. This indicates that the global mean halosteric drift correction leads to an underestimation of the regional halosteric sea level change in the North Atlantic. Subsequently, this means, that the halosteric bias correction overestimates the halosteric sea level change in other regions.

Furthermore, a recent study[26] showed that estimated dynamic ocean mass changes from GRACE (GRACE minus GRD-induced ocean mass change) show trends in the North and South Atlantic that oppose the dynamic ocean mass trends from reanalysis models (Fig. A1 in Carmargo et al.[26]). The reanalysis models reveal a positive dynamic

ocean mass change in the North Atlantic and negative dynamic change in the South Atlantic[26] over the GRACE period, which could potentially account for the substantial difference between the North and South Atlantic regions in GRACE data and the discrepancy to steric-corrected Altimetry.

Another source of uncertainty is the removal of mass changes associated with glacial isostatic adjustment (GIA) and related solid earth deformation[3,17,50]. Despite the application of the same GIA model[51] for both GRACE and altimetry, the GIA-impact on GRACE observations is 4–5 times larger than for altimetry[52]. Consequently, the selection of the GIA model can significantly impact GRACE observations, particularly in the North Atlantic region where variations among GIA models are apparent[17] due to their proximity to former ice masses. This discrepancy among GIA models contributes to an increased GIA-related uncertainty that is not accounted for in the inherent uncertainty of GRACE observations.

In the Indian Ocean, the thermosteric sea level showed a significant increase until 2016 (2.72 mm y⁻¹), whereafter a decreasing thermosteric sea level change is observed (−0.65 mm y⁻¹) (Supplementary Fig. 2). This sudden change correlates with the change in ENSO in 2016, but previous La Niña phases have only temporarily slowed down the warming of the Indian Ocean (Supplementary Fig. 2). Even though a slowing from 4.45 mm y⁻¹ to 3.76 mm y⁻¹ is also observed by altimetry (Supplementary Fig. 6), GRACE and OMrecon (Fig. 3) do not show the accelerated mass change necessary to support the abrupt slowdown in the thermosteric sea level change in the Indian Ocean, implying that this difference may be due to measurement errors in the in-situ observing system.

## Discussion

Ocean mass has been reconstructed from five sources of land-to-ocean water fluxes and compared to 20 years of ocean mass observations from GRACE-GFO and steric-corrected altimetry. All estimates and contributions are provided on a monthly 0.5° grid, allowing assessment of both temporal and spatial ocean mass and sea level change. The three estimates consistently show a significant increase in ocean mass across all regions. Giving equal weight to the three estimates, the global ocean mass has increased at a rate of $2.21 \pm 0.25$ mm y$^{-1}$ water-level equivalent over 2003–2022. While land ice loss and human-caused LWS change are the only contributors to the observed ocean mass trend, inter-annual and seasonal variability is dominated by variations in LWS[53,54].

Our results imply that GFO accurately observes global and regional ocean mass change. The reported lack of closure between steric-corrected altimetry (computed using the standard MWR WTC) and GRACE-GFO[22,23,29,30] is more likely caused by inadequate calibration of the altimeter MWR at the precision required for global sea level budget analysis or global sensitivity to the choice of GIA-solution[18,23,30] rather than technical or data-processing issues of GFO[15]. These GFO speculations have been aided by salinity drift errors in the ARGO measurement system[55], which have led to too high altimetry-based ocean mass changes. Although the OMrecon estimate includes extrapolated mass balance estimates (24 months for the Antarctic and 54 months for glaciers), updated estimates are unlikely to cause a significant imbalance in the budget, which is supported by the long-term agreement between steric-corrected altimetry and GRACE-GFO.

The global ocean mass reconstruction yields good agreement with GRACE-GFO over long-term and seasonal time scales, while some differences are evident over interannual time scales (Figs. 1, 2). Discrepancies are larger regionally, in particular in the Indian Ocean and the Atlantic (Fig. 3), in line with previous research[2,17,26]. Nevertheless, our results show that the majority of observed ocean-mass changes can be appropriately accounted for using land-water and ice models. Interannual variability is dominated by changes in natural land water storage, which shows a negligible trend over the long-term. Instead, our results show that over the last two decades, ice wastage from glaciers and ice sheets, as well as changes resulting from human management of water resources are the main drivers of both global and regional ocean mass change.

This study has revealed that the recent apparent slowdown in ocean mass is likely a result of a prolonged negative ENSO phase since 2020, leading to increased precipitation over land areas and a temporary transfer of mass from the ocean to the land. Additionally, an observed expansion of the Antarctic Ice Sheet contributes also to a decrease in ocean mass change. Reanalysis models indicate that this shift towards Antarctic Ice Sheet growth is driven by accelerated surface mass balance increase from 2020–2022, reaching a historic high in 2022[56]. However, its connection to ENSO is yet to be determined. Nevertheless, we anticipate that this natural variability is temporary, and the El Niño conditions observed in 2023 are expected to prompt a resumption of the long-term acceleration[57] of ocean mass and sea-level.

## Methods

### Ocean mass reconstruction (OMrecon)

OMrecon is the ocean mass reconstructed from three individual assessments of time-varying changes of ice (Greenland Ice Sheet, Antarctic Ice Sheet, and Glaciers) and one of non-glacial land water storage. A number of individual processing steps are necessary to unify the contribution into 0.5° × 0.5° monthly gridded products.

### Glaciers and land water storage

A global glacial model is used to compute gridded monthly glacial mass change. Model outputs are available until March 2018 for the Southern Hemisphere and September 2018 for the Northern Hemisphere[14]. The version used here for glacial ice change is driven by ERA5-reanalysis data from the European Center for Medium-Range Weather Forecasts (ECWMF)[46]. The glacial model has a global (excl. peripheral glaciers) average mass change from 2003 to 2018 of −328 Gt y$^{-1}$, while observation-based gridded mass change estimates[13] show a lower value of −230 Gt y$^{-1}$. The inclusion of marine-terminating glaciers in the glacial model, where some of the ice is already stored below the water surface and a smaller calibration sample used in the model, and thus more likely affected by sampling uncertainties. We, therefore, scale the mass change of the glacial model with a factor of 0.71 to align with the observation-based mass change from ref. 13.

Non-glacial land water storage (LWS) is obtained from WaterGAP2.2e[41] (Water Global Assessment and Prognosis version 2.2e). WaterGAP is a hydrological model, which in this study uses the ERA5-reanalysis[46] as climate forcing. WaterGAP2.2e is an extension of WaterGAP2.2d[45] and runs until December 2022. Human management of water resources (LWS Human), which includes water retention in man-made reservoirs and human water use, that can trigger groundwater depletion. A neutralized variant, where there is no human-induced water management, is used to separate the LWS change into natural LWS change (LWS Natural) and LWS Human.

Mass balance estimates for glaciers are extended in time to December 2022 by adding detrended (trend from 2019–2022 removed) and non-seasonal GFO observations (gE) to a seasonal decomposition of the time series (for each grid cell):

$$vE(t) = \beta + \alpha\,t + A\,\sin(\omega\,t) + gE(t) \qquad (1)$$

vE is the extended mass balance, β, α, A and ω, is the intercept, the linear trend, amplitude, and phase of the modeled mass balance from 2013 to the end of the model coverage and t is the extension time step. We thereby apply the interannual variability from GFO observations, but avoid any effects of a potential linear drift of GFO.

LWS and glacier mass change are present in the same regions and the 300–500 km resolution of GRACE means it often observes both changes simultaneously[58]. To separate LWS from glaciers, the GRACE observed mass balance is convoluted with a filter kernel prior to the above-mentioned extension (Eq. 1). The kernel conserves the mass (i.e., has the sum of 1), but amplifies glacial grid points, while it reduces the mass change signal in the surroundings. The filter kernel is designed so that GRACE observations[20] at glacial grid points reflect the non-seasonal change of modeled glacial mass balance estimates[14] between 2003 and 2018.

This approach effectively seperates glacial mass change observed by GRACE from the observed LWS. The resulting estimate for GRACE-derived LWS, obtained by subtracting the glacial extrapolation from GRACE, demonstrates good agreement with the LWS-model[45] (Supplementary Fig. 5). This lends confidence to the filtering method as it successfully simulates the LWS and glacial mass balance (Supplementary Fig. 5).

The mass balance extension (Eq. 1) is added to the modeled glacier mass balance from 04/2018 in the Southern Hemisphere (57 months), 09/2018 in the Northern Hemisphere (52 months) to simulate the mass balance until the end of 2022.

Gridded glacial mass-balance estimates are not yet available to validate the reconstructed glacial mass balance from 2019 to 2022. The World Glacier Monitoring Service (WGMS) has released preliminary regional estimates[59] for 2019–2022 showing a global average of −320 ± 58 Gt y$^{-1}$ (excluding pheripheral Antarctic and Greenland glaciers). The glacial reconstruction used here yields a similar average global mass balance of −306 ± 60 Gt y$^{-1}$ from 2019–2022.

The uncertainty of the annual average is used and an extra 20% is added to account for the additional uncertainty originating from the extension.

## Antarctic ice sheet

The mass change estimate of the Antarctic is based on the IMBIE multi-method assessment[12] and normalized spatial mass changes and seasonal mass change derived from GRACE. The IMBIE estimate is region-averaged with no seasonal component and is a weighted average of mass balance estimates from altimetry, GRACE, and input-output methods. Altimetry estimates do not include the peripheral glaciers, but estimates from GRACE and some of the input-output methods do. To account for this, the method from ref. [13] is applied, where half of the mass balance from Antarctic glaciers[13] is added to the IMBIE assessment divided equally between the 3 regions (West, East and Antarctic Peninsula). The resulting total contribution of Antarctic peripheral glaciers is varying between −5 and −15 Gt y$^{-1}$ equivalent to 0.01–0.04 mm y$^{-1}$ sea level change.

To distribute the region-averaged mass changes over the Antarctic, we convolute the IMBIE estimates with normalized GRACE-derived[20] mass changes over the Antarctic. The gridded average seasonal change of GRACE is hereafter added to the gridded mass changes from the convoluted IMBIE-estimates.

For the period from 01/2021 to 12/2022 (24 months), the Antarctic mass balance is extended utilizing GRACE-FO observations in the same way as glaciers and land water storage (Eq. 1).

## Greenland ice sheet

The Geological Survey of Denmark and Greenland (GEUS) provides daily updated mass balance estimates dating back to 1840[40]. These estimates are reported as averages for each of the Zwally drainage basins and are extrapolated to each grid cell in a 0.5° grid and monthly averaged. The referenced mass balance estimate for the Greenland Ice Sheet does not include peripheral glaciers but is added from the glacial model[14] to provide the total mass balance estimate of Greenland. GRACE measurement of Greenland includes peripheral glaciers due to the coarse spatial resolution of GRACE. The difference between the ice sheet mass balance and GRACE-GFO, is utilized as the GRACE-GFO measured peripheral glacier estimate (gE in Eq. 1). The same methodology as for glaciers is then used to extend the mass balance of peripheral glaciers using Eq. 1.

## GRD-induced sea level change

The ocean mass reconstruction is derived from the Gravitational, Rotational, and Deformational (GRD) response from each loading (Greenland, Antarctic, Glaciers, and TWS) using the ISSM-SEESAW model framework[60]. For each of the four land-water contributions, 1000 ensembles are constructed assuming a Gaussian distribution of the measurement uncertainty associated with the land/ice mass change estimate. The GRD-induced mass change from the ensemble members is computed for each month with and without rotational feedback. Following recent recommendations[42], the seasonal fingerprint does not include rotational feedback when compared to GRACE and altimetry products, since the effects of polar motion, Chandler wobble and pole tides have been removed from those products[42].

Interpolation from the original grid to the triangular mesh grid used by ISSM-SEESAW and further interpolation to the 0.5° grid used in this study can lead to small changes in the total mass due to changes in ocean area. A correction factor is applied to ensure that mass is conserved so that the land water equivalent mass loss is equal to the mass of the induced relative sea level change of the final 0.5° grid. We assumed that 361.8 Gt of land mass loss equals 1 mm of whole-ocean sea level rise (equivalent to an area of 361,800,000 km$^2$). Please note, that the ocean mask throughout this study (Fig. 3), is different from the global ocean, and has an area of 299,900,000 km$^2$. The final ensemble means and spread determines the central estimate (Supplementary Fig. 4) and uncertainty (1 sigma).

From the GRD calculations, both vertical solid-earth deformation (used for calculating ocean bottom deformation), relative sea level

(used for mass comparisons), and absolute sea level are provided in a monthly 0.5-degree grid for each source.

The mean and standard deviation of the ensemble constitutes the central estimate (Supplementary Fig. 1a−c) and uncertainty of the ocean mass contribution of each source (Supplementary Fig. 2a−c). The four sources are summed together and corrected for global mean atmospheric water mass to form the ocean mass reconstruction (Supplementary Fig. 1d), which can be compared to GRACE ocean mass observations (Supplementary Fig. 1e) and steric-corrected altimetry (Supplementary Fig. 1f).

## GRACE-observations

GRACE gravitational measurements are used to extend land and ice mass balance estimates and are used as an estimate of ocean mass. The estimations rely on the same GRACE mascon product (GSFC mascons RL06)[20], where the global average surface pressure from an atmospheric and non-tidal ocean model (GAD) has been removed over ocean grid cells. Mass changes related to glacial isostatic adjustment[33] (GIA) have been removed using the GIA ICE6G-D model[61].

The native temporal resolution of GRACE varies from 20-35 days, having the central day around mid-month (typically between day 13 and day 20 of the month). To unify estimates, the gridded GRACE estimates are interpolated to mid-month for months where observations exist 10 days before or after the mid-month. Missing GRACE months have been interpolated, after removing the seasonal component, which is restored after interpolation. We keep the 13-month gap between GRACE and GRACE Follow-On.

## Ocean mass from steric-corrected altimetry

Steric-corrected altimetry is obtained by subtracting the steric sea level heights from altimetry-derived sea surface height anomalies.

## Sea surface heights from altimetry

Along track sea level anomalies (SLA) with the default MWR WTC applied from Jason-1 (2003-01-01–2008-07-12), Jason-2 (2008-07-13–2016-03-17), Jason-3 (2016-03-18–2022-04-15) and Sentinel-6 Michael Freilich (2022-04-15–2022-12-31) are obtained from the Radar Altimeter Database System[62]. An intermission bias is removed from each following satellite, by calculating the mean difference of up to 37 cycles (~1 year) where the satellites are flying in tandem missions. The MWR WTC is obtained in the same way as SLA and reapplied to the SLA record. MWR WTC from 2003-01-01 to 2016-03-17 is combined with WTC calculated from climate data records[39], following the approach of ref. [38], from 2016-03-17 to 2022-12-31. The MWR/CDR combined WTC is applied to the uncorrected sea level anomalies, which are linearly interpolated in a 0.5-degree grid using all observations of each calendar month. The uncertainty is estimated from the standard deviation of all the observations of monthly 3 × 3 degree cells ($\sigma_{grid}$) combined with the standard deviation of the intermission bias ($\sigma_{IMB}$), which is uncorrelated to ($\sigma_{grid}$).

$$\sigma_{SLA} = \sqrt{\sigma_{grid}^2 + \sigma_{IMB}^2} \tag{2}$$

Sea level observations from satellite altimetry are observed in a geocentric frame with reference to a reference ellipsoid or mean sea surface above the reference ellipsoid, while GRACE observes the mass change of the ocean column. Hence, is the ocean bottom deformation[31,34], i.e., the elastic solid-earth deformation caused by loading change observed by GRACE, but undetected by altimetry. Therefore, we remove ocean bottom deformation from the altimetry observed sea surface height anomalies. Furthermore, since GIA effects are removed from GRACE, we also remove the spatial-varying absolute sea level change caused by GIA[33] from altimetry, following ref. [50]. The variance between the used GIA model[33] and another GIA

model[63] is applied as the GIA-uncertainty for steric-corrected Altimetry.

**Steric sea level.** Steric sea level change resulting from changes in salinity and temperature are not changing the ocean mass and are hence removed from the altimetric sea surface heights. Thermosteric and halosteric sea level change is based on monthly updated, reanalyzed and gridded temperature and salinity estimates[64,65] calculated between 0-5400 meters using the HOMAGE software (https://github.com/podaac/HOMaGE)[66], which is built upon the TEOS-10 ocean software package (http://www.teos-10.org). The global mean halosteric sea level change should globally be near zero[67] but varies spatially and seasonally. However, issues with conductivity calibration drift in the ARGO-measurement system have been observed since 2015[36,37] causing an unnatural anomalous halosteric sea level change, that manifests globally (Fig. 1 and Supplementary Fig. 6). To account for this, we remove the global lowpass filtered change, but keep frequencies of less than 1 year.

In-situ observations of below 2000 meters are sparse and the sea level contribution is generally difficult to assess[68,69]. Limiting the reanalysis data[64] to depths above 2000 meters, lowers the global steric sea level trend by $0.10 \pm 0.04$ mm y$^{-1}$. Thus, the deep steric estimate is consistent with the commonly applied deep steric contribution estimate[35] of $0.12 \pm 0.03$ mm y$^{-1}$. Regionally the deep steric contribution varies between 0.00–0.29 mm y$^{-1}$.

### Dynamic mass change and ocean mask
Dynamic ocean mass change is included in both steric-corrected altimetry and GRACE are estimates of manometric sea level change[67], but not in OMrecon. Globally, dynamic ocean mass change has a zero-mean mass contribution[67], but is significant over local scales as water mass is shifted from one location to another[70]. This effect is, in particular, large in coastal shallow regions, where the dynamic sea level change explains the major part of recent sea level changes measured by tide gauges[70]. By choosing a narrow 50 km coastal cut-off across all estimates (compared to the commonly used 200–500 km coastal cutoff[4,16,23,25,29]), most of the dynamic sea level remains in the estimates, thereby reducing bias from mass flux between the open ocean and coastal regions not recovered in OMrecon. However, very shallow ocean regions (maximum depth of less than 200 meters) are masked during the analysis given sparse observations by Argo and challenges associated with coastal altimetry. Furthermore, we mask areas surrounding the two large megathrust earthquakes (Sumatra in 2004 and Sendai in 2011), which have a visible impact on GRACE observations.

### Trend, phase, and amplitude estimation
We use bootstrapping principles to estimate the trend, phase, and amplitude. Each time series is weighted by the inversed-squared error. From the weighted time series, a random selection is drawn, with a sample size equal in size to the time series length. From the time series sample, the trend, phase, and amplitude are calculated. This is repeated for 10,000 samples and from the mean and standard deviation of the distribution we estimate the mean trend, phase, and amplitude, and associated uncertainty as shown in Table 1.

## Data availability
All data needed to support the findings in this manuscript are provided in the source data file 'SourceData.xlsx'. Monthly gridded contributions to ocean mass change, as well as the GRACE ocean mass estimates, altimetry sea surface heights (both MWR and MWR/CDR), and steric sea level are available in the public repository[71]. Greenland mass change[72], natural land water storage[73], human land water storage[74], IMBIE Antarctic mass change[75], and glacier-specific mass change[76] are directly available from respective sources. Source data are provided with this paper.

## Code availability
GRD effects from loading changes have been computed with the publicly available Ice-sheet and Sea-level System Model (ISSM)[60]. Scripts used for generating the plots in this paper are available from the corresponding author upon reasonable request.

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

## Acknowledgements

C.B.L. was funded by the Carlsberg Foundation (Grant Number CF22-0146). M.A.K. and C.W. are funded by the Australian Research Council Special Research Initiative, Australian Center for Excellence in Antarctic Science (Project Number SR200100008).

## Author contributions

The study was conceptualized by CBL and OBA. CBL developed the required methodologies, data computation, and result analysis. The results were thoroughly discussed by CBL, OBA, CW, and MK. CBL authored the manuscript, with contributions from OBA, CW, and MK. Modeled glacier and land water storage data and their relevance for this study were provided and discussed by BM, JHM, HMS, and PD. CBL, OBA, BM, JHM, HMS, PD, CW, and MK contributed with comments to improve the manuscript.

## Competing interests

The authors declare no competing interests.
