## [Peer Review File · Nature Communications]

REVIEWER COMMENTS

Reviewer #1 (Remarks to the Author):

The paper 'Global and regional ocean mass budget closure since 2003' by Ludwigsen et al. provides a comprehensive update of the global and regional sea level budget, its drivers and contributors. It provides important new insights into the state of the sea level observing systems, and attributes recent sea level signals to climate signals and instrument trends. The paper is an important and impactful contribution to the field, and I recommend publication after my comments below are addressed.

Line 41-44

This may not always be true everywhere. For some regions, ocean dynamics could play an outsize role; vertical land motion plays a major role along many coastlines. While that is relative sea level (as opposed to the purely geocentric view), it is relative sea level that matters when describing risks and vulnerabilities. This should be corrected for clarity.

Line 79-80

The comment about global atmospheric mass change is a bit odd here - not clear what the purpose is. The reference to Chen et al. is also somewhat odd - that paper addressed a processing 'accounting' detail for G/GFO and background models for deg 0, as opposed to addressing the global water cycle as such.

Line 124

What is the source of the global mean atmosphere mass here?

Line 179-181

The attribution of the differences is somewhat ad hoc - is there more quantitative evidence that the precipitation under-estimates are consistent with the extra mass over land?

Line 185-189

This phase-offset was discussed in ref35 - it would be appropriate to mention it here again (eg, in line 189)

Line 210-212

Does the NAO-correlation pertain to both the higher/lower Greenland mass loss rates, or just the 2019-2020 increase? Please clarify.

Line 238-241

If I understood correctly, the halosteric drift correction can only be estimated in a global-mean sense. Any regional correction with this global mean may then over or underestimate the true regional halosteric bias, is that correct? It would be helpful to spell this out a little more clearly.

Line 243-249

I have looked up ref32 - I did not find statements that would support the claim here that 'dynamic ocean mass changes from GRACE (GRACE minus GRD-induced ocean mass change) is generally underestimated compared to dynamic ocean mass change from reanalysis models'. Ref32 does mention some challenges in coastal areas, and the South Atlantic, but this may just as well be a bias in the reanalysis models. Please reword to reflect the findings of ref32 (e.g., their Fig.1) properly.

Line 283-284

The dominant role of LWS on inter annual time scales has previously been highlighted, please add references accordingly.

Line 304

I'm not sure there is strong evidence to claim that the inter annual LWS signal is 'natural' - it implies that this is the response of the unperturbed climate system. But it may still contain a significant 'forced' component, e.g., if human activity related to water use is also correlated with ENSO (that could be the case when irrigation and water retention is correlated with ENSO and water cycle and temperature patterns).

Line 312-313

The return to higher ocean mass trends is an interesting hypothesis, but speculative at this point. Thus, 'leading' should perhaps be re-phrased into 'would lead'?

Method Section:

Line 9-20

What are the observations for glacier mass change based on? If it includes G/GFO, then the scaling applied here would introduce a dependency of OMrecon on G/GFO - is this the case? If so, it needs to be discussed.

Line 50

Wording: 'segregates' -> 'separates'

Line 166

'Contrary to GRACE' - not sure I understand this: G/GFO also measure in the geocentric frame. Subsequently, a geocenter correction/component is applied. So 'contrary' is not quite the right way to describe & contrast the altimetric and gravimetry observing systems?

Line 193

I'm curious: have you assessed the impact of using a coastal mask of 50 vs 300 km in your estimates? I also strongly suggest that you include the land-sea mask in your data repository.

Line 208-214

Can you add some references here please? Did you use any open-source tools?

Reviewer #2 (Remarks to the Author):

Review of Global and regional ocean mass Budget closure since 2003, by Ludwigsen et al.

This Work presents several time series of ocean mass from different sources from 2003 to 2022. These sources are gravimetry measurements from GRACE and its continuation GFO, reconstruction of changes in land water and ice (OMrecon), and the estimation of the ocean mass changes from altimetry data corrected for the steric contribution. The OMrecon includes estimations of the contributions from Greenland, Antarctica, Glaciers, human and natural water storage. The sea level estimated from altimetry also includes corrections for the water vapour in the atmosphere, as well of GIA corrections. In my opinion this is a very complete and exhaustive work. The match between the different sources within the uncertainty level (Budget closure) allows the authors to establish in a reliable way the change that the mass of water in the oceans have experienced since 2003 on a global scale and to identify its sources. Beside this, the authors also analyse these changes on a regional scale and present a description of the seasonal cycle of the ocean water mass.

Therefore, I think that this work merits publication. However, there are some questions that I have not understood very well and I would like that the authors clarify them previous to publication. Find below these doubts and concerns.

First, English is not my mother tongue, and therefore I do not feel qualified to review the English. Nevertheless, some paragraphs have been difficult to understand for me and I have had to read them several times. I think that the redaction should be carefully reviewed.

Line 51. Figure 1 is mentioned for the first time. In the legend of figure 1 I can see a curve for OMrecon, but this time series and the meaning of this acronym has not been defined at this moment. In fact, it is not defined until line 112. So, I am looking at OMrecon time series and trying to understand what it is until line 112. This should be explained before the reader is referred to it.

Line 92. "...removing the annual-mean halosteric change." Then, at the end of this line, the authors use the expression: "steric-corrected". I do not understand why the authors do not simply remove the steric contribution, including both the thermosteric and the halosteric contributions. When they say that they remove the halosteric change, it seems to the reader (at least to me), that they do not remove the thermosteric one. But then they use the term steric-corrected, and I do not know if now the thermosteric contribution has also been calculated. On a global scale it is the thermosteric contribution the one with a major role.

Here I simply have a doubt. Can the ocean bottom deformation alter the mass of the ocean? I understand that it can produce changes of sea level on a regional scale, but the mass of water should not change.

Line 95. Figure 1 is mentioned again. As I said, I do not understand the legend of this figure that should be re-made. For instance, in figure 1b, which is a zoom of figure 1a, in the legend it is indicated that black line is for GRACE, dashed and dotted magenta lines for uncorrected and corrected altimetry. But I see three magenta lines. There is also a shaded range that is not explained in the legend.

Lines 105-106. The term steric-corrected includes both the thermosteric and halosteric contributions. Then, why do you mention “...the remaining of the halosteric contribution”?

Line 124. You say that you subtract “global mean atmospheric mass” I suppose you mean: global mean atmospheric mass change. Here I have another doubt: The only cause of changes in atmosphere mass is a change in water vapour?

Table 1. I think that the sign criterion should be reviewed. All the changes are positive. I understand that the change of mass of the atmosphere produces a decrease of the mass of water in the oceans. That is the reason for subtracting it to the barystatic component. Therefore, I think that those changes that produce an increase of the mass of water in the oceans should be included in table 1 as positive, and those producing a decrease should be included as negative.

When I sum all the contributions (barystatic), I obtain 2.21 mm/yr instead of 2.24. Then the barystatic minus atmosphere is also different from the one in the table.

I have one doubt concerning the influence of human land water storage. If I am not wrong, this is included as a positive contribution to the total mass of water in the ocean. Does it mean that there is less water in the artificial reservoirs? I guess this is because of a larger consume.

Lines 153-154. GRACE show a global ocean-mass change of 38.1 mm. But in table 1, the GRACE change from 2003 to 2022 is 2.10 mm/yr The length of the period is 20 years. Then $2.1 \times 20 = 42$ mm instead of 38.1. Other figures do not seem to be right, or maybe I am missing something.

Lines 169-171. The authors say that OMrecon deviates from GRACE and steric-corrected altimetry for the last 3 years of the time series (2020-2022), but I see similar differences in figure 1b for the period 2015-2017.

Furthermore, the authors hypothesize on the effect of El niño and la niña on these changes, but 2020-2022 is a period when data have been extrapolated. Extrapolation almost always is wrong. This is a much simpler explanation for the differences between different data sources.

Line 187. “It is evident that the seasonal signal is dominated by natural land water storage”. This is also stated in lines 284-285. But, when I look at figure 1d, I see that the amplitude of Greenland (blue) and Antarctica (yellow) seasonal cycles are larger than that of the LWS. It is true that both are almost in phase opposition, then they could counter-balance, but, when the seasonal cycle of Antarctica is zero, the Greenland one is in its minimum value. Hence, I understand that the ice sheets of Greenland and Antarctica make an important contribution to the seasonal variability.

Reviewer #3 (Remarks to the Author):

Since 2016, the sea-level budget (sea level(altimetry) = mass(gravimetry) + steric(argo)) has evaded closure, with altimetry estimating significantly more global sea level rise compared to gravimetry + argo.

The manuscript “Global and regional ocean mass budget closure since 2003” submitted to Nature Communications attempts to close the ocean mass budget and investigate the causes of ocean mass budget misclosure. Ocean mass budget misclosure is coincident with significant flattening of the GRACE/GRACE-FO ocean mass curve during a period of mission-gaps and degraded GRACE/GRACE-FO ACC data products. This is the motivation for creating an additional “independent” ocean mass reconstruction (OMrecon) to determine whether GRACE/GRACE-FO models capture the full extent of ocean mass increase post-2016.

By correcting Jason-3 altimetry for an adjusted wet troposphere correction (WTC) and argo for salinity drifts ocean mass misclosure is reduced although still remains for some ocean basins and at interannual scales which is also consistent with recent work (e.g. Barnoud et al. 2023). It is not obvious to me what about these findings are particularly new or novel. Previous attempts to close the ocean mass budget have achieved closure to within a similar degree and previous work has already highlighted that misclosure is prominent regionally and at interannual timescales. I can however, appreciate the work that has gone into producing the OMrecon, which provides a mostly-independent dataset to compare to GRACE/GRACE-FO. For the most part, their reconstruction matches well the GRACE/GRACE-FO observed ocean mass change, supporting their conclusion that GRACE/GRACE-FO is not the source of misclosure.

Summary

The summary needs a rewrite, there are quite a few typos and confusing sentences. I think it is important to highlight that closure of the global ocean mass budget has been a problem but near-closure has already been achieved by accounting for argo biases and WTC from alternative MWR (e.g. Barnoud et al., 2023). However, misclosure of the global ocean mass budget remains at interannual scales and in individual ocean basins.

Main

1. OMrecon is referred to as an independent dataset of ocean mass change throughout the paper. I take issue with this because GRACE/GRACE-FO data are used to generate OMrecon in multiple ways by (1) using GRACE-FO interannual variability to extend watergap, (2) using IMBIE combined assessment for Antarctic mass change, and (3) extending the mass balance of peripheral glaciers.
2. Paragraph starting line 59 needs a rewrite, split the explanation of how ocean mass changes can be estimated from altimetry and ocean mass reconstructions into separate sentences. Give examples of where these land contributions come from (e.g. land surface models?). My first thought reading this was that these land contributions come from GRACE/GRACE-FO and therefore, this does not provide an “independent estimate”
3. Paragraph starting line 97: you reduce the misclosure of the ocean mass budget by correcting altimetry for the MWR WTC and correcting argo for an “annual global halosteric contribution”. Barnoud et al., 2023 also replaced the argo-based thermosteric component, would this account for the remaining gap in altimetry-steric and GRACE/GFO?
4. Figure 1: I find this figure really difficult to understand, some colours are used multiple times for different data. I also think it would be appropriate to interpolate between missing GRACE months (leaving the mission gap empty). Tick marks on the left side of the subplots would also help
5. Paragraph starting 169: add numbers to support the statement that “The contrasting pattern can be attributed to the tendencies of the climate reanalysis data...In particular for Africa”. Add

Australia to ext Fig 1. If the climate reanalysis data underestimates LWS during la nina this will be very obvious compared to the GRACE-FO data

6. Line 190-192: Why consider the difference in OMrecon with/without global mean atmospheric mass?
7. Line 203: Looking at Fig 2, I'm not convinced of this relationship, the largest deviations between OMrecon and GRACE/altimetry seem to occur before large la nina or el nino periods. Could you provide some statistics to back up your statement or add OMrecon – GRACE to Fig 1a so we can line up where the greatest differences occur with respect to ENSO?
8. Line 230: "it is worth noting that there are contributing factors that are also relevant to highlight" what are they?
9. Line 240-241: I'm confused by this statement

Conclusions

line 291-292: This is misleading. GRACE-FO accelerometer errors will not cause drifts in the ocean mass estimates. Biases on the accelerometer measurements are accounted for during gravity field inversion.

Minor:

Line 94: which steric corrected altimetry, there are three plotted in Figure 1? And which subplot of Figure 1?

Line 102: extra bracket needs to be removed

Line 103: you haven't closed the post-2016 budget yet. How much is the "small divergent ocean mass change still evident in the GFO era"?

Line 105: "CDR" referenced before definition

Line 110: two uses of "observed"

Line 110: I think you need to specify in the main text where these time series of land-based water mass change are coming from

Line 120: the IMBIE estimates terminate in December 2020

Line 188: add ref to Fig 1d

Line 190: while a -> while

Line 301: regionally -> regional

Supplementary

1. GACE-Observations
 - a. What is GAD
 - b. Isn't GAD removed just from ocean mascons anywhere? Therefore, you aren't using two different solutions.
 - c. Explicitly say whose solution you are using (GSFC mascons RL06?)
 - d. Specify the GIA model in text (ICE6G_D)
 - e. I think it would be appropriate to interpolate over missing GRACE months

REVIEWER COMMENTS

Reviewer #1 (Remarks to the Author):

The paper 'Global and regional ocean mass budget closure since 2003' by Ludwigsen et al. provides a comprehensive update of the global and regional sea level budget, its drivers and contributors. It provides important new insights into the state of the sea level observing systems, and attributes recent sea level signals to climate signals and instrument trends. The paper is an important and impactful contribution to the field, and I recommend publication after my comments below are addressed.

Line 41-44

This may not always be true everywhere. For some regions, ocean dynamics could play an outside role; vertical land motion plays a major role along many coastlines. While that is relative sea level (as opposed to the purely geocentric view), it is relative sea level that matters when describing risks and vulnerabilities. This should be corrected for clarity.

True. We changed the wording so that it only says 'most ocean regions'.

Line 79-80

The comment about global atmospheric mass change is a bit odd here - not clear what the purpose is. The reference to Chen et al. is also somewhat odd - that paper addressed a processing 'accounting' detail for G/GFO and background models for deg 0, as opposed to addressing the global water cycle as such.

The sentence in the parenthesis and reference has been removed.

Line 124

What is the source of the global mean atmosphere mass here?

Reference is added (GAA-product of GRACE Tellus Rel. 06).

Line 179-181

The attribution of the differences is somewhat ad hoc - is there more quantitative evidence that the precipitation under-estimates are consistent with the extra mass over land?

A sub-figure is added to suppl. Fig. 2 (previous ext. Fig. 1), which shows, that the total modelled LWS agrees with GRACE-observed LWS, when excluding the mentioned regions over Africa.

Line 185-189

This phase-offset was discussed in ref35 - it would be appropriate to mention it here again (e.g., in line 189)

Reference is added.

Line 210-212

Does the NAO-correlation pertain to both the higher/lower Greenland mass loss rates, or just the 2019-2020 increase? Please clarify.

We have rephrased the paragraph. Both high and low Greenland rates are correlated to NAO.

Line 238-241

If I understood correctly, the halosteric drift correction can only be estimated in a global-mean sense. Any regional correction with this global mean may then over or underestimate the true regional halosteric bias, is that correct? It would be helpful to spell this out a little more clearly.

This is correctly understood. We have rephrased the paragraph to clarify this.

Line 243-249

I have looked up ref32 - I did not find statements that would support the claim here that 'dynamic ocean mass changes from GRACE (GRACE minus GRD-induced ocean mass change) is generally underestimated compared to dynamic ocean mass change from reanalysis models'. Ref32 does mention some challenges in coastal areas, and the South Atlantic, but this may just as well be a bias in the reanalysis models. Please reword to reflect the findings of ref32 (e.g., their Fig.1) properly.

We refer to Figure A1 in ref32. However, 'generally underestimated', might be an over-interpretation of the figure. More correct is that GRACE-estimated dynamic sea level shows opposite trends in the Atlantic compared to the reanalysis model. We have edited the text accordingly.

Line 283-284

The dominant role of LWS on inter-annual time scales has previously been highlighted, please add references accordingly.

References added.

Line 304

I'm not sure there is strong evidence to claim that the inter annual LWS signal is 'natural' - it implies that this is the response of the unperturbed climate system. But it may still contain a significant 'forced' component, e.g., if human activity related to water use is also correlated with ENSO (that could be the case when irrigation and water retention is correlated with ENSO and water cycle and temperature patterns).

From Figure 2 it is evident that the interannual variation of OMrecon follows the natural LWS. While a correlation between ENSO and LWS Human is visible (i.e. El Nino 2016), the variation is small compared to the natural variation of LWS. Below is LWS natural from Figure 2 plotted on the same y-axis as human-caused LWS change.

Line 312-313

The return to higher ocean mass trends is an interesting hypothesis, but speculative at this point. Thus, 'leading' should perhaps be re-phrased into 'would lead'?

Agree, we softened the language a little.

Method Section:

Line 9-20

What are the observations for glacier mass change based on? If it includes G/GFO, then the scaling applied here would introduce a dependency of OMrecon on G/GFO - is this the case? If so, it needs to be discussed.

Both the glacial mass model and the observations from Hugonnet et al are independent of GRACE observations.

Line 50

Wording: 'segregates' -> 'separates'

Changed

Line 166

'Contrary to GRACE' - not sure I understand this: G/GFO also measure in the geocentric frame. Subsequently, a geocenter correction/component is applied. So 'contrary' is not quite the right way to describe & contrast the altimetric and gravimetry observing systems?

Agree, that referring to the geocentric frame is the wrong way to highlight the difference between GRACE and altimetry-derived ocean mass observations. We changed the text, so it says that GRACE observes the mass change of the ocean column, while altimetry observes the sea level change w.r.t. the reference ellipsoid and thus is not able to separate ocean bottom deformation from sea level change.

Line 193

I'm curious: have you assessed the impact of using a coastal mask of 50 vs 300 km in your estimates? I also strongly suggest that you include the land-sea mask in your data repository.

Yes, the impact is minor. In extended table 1. we have added trend estimates for different ocean masks.

Line 208-214

Can you add some references here please? Did you use any open-source tools?

Software references have been added.

Reviewer #2 (Remarks to the Author):

Review of Global and regional ocean mass Budget closure since 2003, by Ludwigsen et al.

This Work presents several time series of ocean mass from different sources from 2003 to 2022. These sources are gravimetry measurements from GRACE and its continuation GFO, reconstruction of changes in land water and ice (OMrecon), and the estimation of the ocean mass changes from altimetry data corrected for the steric contribution. The OMrecon includes estimations of the contributions from Greenland, Antarctica, Glaciers, human and natural water storage. The sea level estimated from altimetry also includes corrections for the water vapour in the atmosphere, as well of GIA corrections.

In my opinion this is a very complete and exhaustive work. The match between the different sources within the uncertainty level (Budget closure) allows the authors to establish in a reliable way the change that the mass of water in the oceans have experienced since 2003 on a global scale and to identify its sources. Beside this, the authors also analyse these changes on a regional scale and present a description of the seasonal cycle of the ocean water mass.

Therefore, I think that this work merits publication. However, there are some questions that I have not understood very well and I would like that the authors clarify them previous to publication. Find below these doubts and concerns.

First, English is not my mother tongue, and therefore I do not feel qualified to review the English. Nevertheless, some paragraphs have been difficult to understand for me and I have had to read them several times. I think that the redaction should be carefully reviewed.

We have reviewed the language throughout the paper.

Line 51. Figure 1 is mentioned for the first time. In the legend of figure 1 I can see a curve for OMrecon, but this time series and the meaning of this acronym has not been defined at this moment. In fact, it is not defined until line 112. So, I am looking at OMrecon time series and trying to understand what it is until line 112. This should be explained before the reader is referred to it.

We have incorporated a new supplement figure (suppl. Fig. 1) that illustrates the GRACE ocean mass time series, highlights the missing months, and showcases the GRACE gap, along with associated trends over five 4-year periods. Additionally, we have included the explanation of 'steric-corrected altimetry' and 'OMrecon' in the section following the previously mentioned GRACE section.

Line 92. "...removing the annual-mean halosteric change." Then, at the end of this line, the authors use the expression: "steric-corrected". I do not understand why the authors do not simply remove the steric contribution, including both the thermosteric and the halosteric contributions. When they say that they remove the halosteric change, it seems to the reader (at least to me), that they do not remove the thermosteric one. But then they use the term steric-corrected, and I do not know if now the thermosteric contribution has also been calculated. On a global scale it is the thermosteric contribution the one with a major role

We have edited the text to clarify our approach to account for the known salinity drift error in the ARGO observing system The halosteric sea level change should ideally make only a near-zero annual contribution to sea level change on a global scale. However, it can exhibit significant regional and seasonal variations. To address this, we have chosen to retain only

the frequencies of halosteric change that are less than one year. These retained frequencies are incorporated into the steric sea level and subsequently removed from the altimetry data, which we then denote steric-corrected altimetry (altimetry minus steric corrected for salinity drift). We believe this revised text offers a clearer explanation of our methodology.

Here I simply have a doubt. Can the ocean bottom deformation alter the mass of the ocean? I understand that it can produce changes of sea level on a regional scale, but the mass of water should not change.

Ocean bottom deformation (OBD) itself does not alter the mass of the ocean; however geocentric altimetry observations measure total volume change, and the volume change associated with both GIA and OBD must be removed to derive accurate ocean mass change. Therefore, in an altimetric observing system, it is essential to correct for both OBD and GIA. Frederikse et al. 2017 and Vishwakarma et al. 2020 provide detailed explanations of this correction.

Line 95. Figure 1 is mentioned again. As I said, I do not understand the legend of this figure that should be re-made. For instance, in figure 1b, which is a zoom of figure 1a, in the legend it is indicated that black line is for GRACE, dashed and dotted magenta lines for uncorrected and corrected altimetry. But I see three magenta lines. There is also a shaded range that is not explained in the legend.

We have made slight changes to the figure while retaining the legends of Figures 1a and 1b (now figure 2). The caption clarifies that the two (dotted and dashed) magenta lines have been added to Figure 1b (and are accordingly included in the legend of Figure B), while the other lines in Figure 1b and lines in Figures 1d and 1e, draw information from the legend of Figure 1a. The caption also notes that the grey vertical shaded area represents the GRACE gaps, and the shaded areas of the lines indicate 1-sigma uncertainties. This approach strikes a balance between avoiding a cluttered plot and providing essential information in the legend.

Lines 105-106. The term steric-corrected includes both the thermosteric and halosteric contributions. Then, why do you mention "...the remaining of the halosteric contribution"? As per the previous comment on line 92, we have made adjustments to the text concerning the correction of the halosteric drift.

Line 124. You say that you subtract "global mean atmospheric mass" I suppose you mean: global mean atmospheric mass change. Here I have another doubt: The only cause of changes in atmosphere mass is a change in water vapour?

True, it is global mean atmospheric mass change. This sentence has however been removed according to reviewer #3's comments.

Table 1. I think that the sign criterion should be reviewed. All the changes are positive. I understand that the change of mass of the atmosphere produces a decrease of the mass of water in the oceans. That is the reason for subtracting it to the barostatic component. Therefore, I think that those changes that produce an increase of the mass of water in the oceans should be included in table 1 as positive, and those producing a decrease should be included as negative.

When I sum all the contributions (barystatic), I obtain 2.21 mm/yr instead of 2.24. Then the barystatic minus atmosphere is also different from the one in the table.

We have changed the sign for atmospheric mass change, and correct - there was also a mistake in the table, which has been corrected.

I have one doubt concerning the influence of human land water storage. If I am not wrong, this is included as a positive contribution to the total mass of water in the ocean. Does it mean that there is less water in the artificial reservoirs? I guess this is because of a larger consume.

Yes, artificial reservoirs and groundwater extraction are the main reasons for human land water storage change, which we think is clear in the text. Hence are no corrections been made.

Lines 153-154. GRACE show a global ocean-mass change of 38.1 mm. But in table 1, the GRACE change from 2003 to 2022 is 2.10 mm/yr. The length of the period is 20 years. Then $2.1 \times 20 = 42$ mm instead of 38.1. Other figures do not seem to be right, or maybe I am missing something.

The numbers were originally computed as the difference between the averages of 3-year periods, specifically 2003-2005 and 2020-2022, which was unintentionally omitted from the manuscript. In response to the reviewer's suggestion, we have recalculated these values by multiplying the trend by 20 years.

Lines 169-171. The authors say that OMrecon deviates from GRACE and steric-corrected altimetry for the last 3 years of the timer series (2020-2022), but I see similar differences in figure 1b for the period 2015-2017.

Furthermore, the authors hypothesize on the effect of El Niño and la Niña on these changes, but 2020-2022 is a period when data have been extrapolated. Extrapolation almost always is wrong. This is a much simpler explanation for the differences between different data sources.

It is indeed true that other periods also exhibit relatively large differences among the three ocean mass estimates. The deviation observed during 2015-2017 coincides with an El Niño event and is therefore in line with our hypothesis. Only glaciers and Antarctic mass balances have been extrapolated. Glaciers does not display significant interannual differences, as shown in Figure 2, while Antarctica makes a significant shift from mass loss to mass gain from after 2020. This is change is in part confirmed by reanalysis models (ref 55), that shows a significant SMB change from 2020-2022. This has been added to the discussion.

Line 187. "It is evident that the seasonal signal is dominated by natural land water storage". This is also stated in lines 284-285. But, when I look at figure 1d, I see that the amplitude of Greenland (blue) and Antarctica (yellow) seasonal cycles are larger than that of the LWS. It is true that both are almost in phase opposition, then they could counter-balance, but, when the seasonal cycle of Antarctica is zero, the Greenland one is in its minimum value. Hence, I understand that the ice sheets of Greenland and Antarctica make an important contribution to the seasonal variability.

We believe the reviewer may have misinterpreted Figure 1d-e, as both figures illustrate the seasonal cycle but differ in the scale of the y-axis. We have now addressed this difference in the figure caption.

Reviewer 3# (Remarks to the Author):

Since 2016, the sea-level budget (sea level(altimetry) = mass(gravimetry) + steric(argo)) has evaded closure, with altimetry estimating significantly more global sea level rise compared to gravimetry + argo. The manuscript "Global and regional ocean mass budget closure since 2003" submitted to Nature Communications attempts to close the ocean mass budget and investigate the causes of ocean mass budget misclosure. Ocean mass budget misclosure is coincident with significant flattening of the GRACE/GRACE-FO ocean mass curve during a period of mission-gaps and degraded GRACE/GRACE-FO ACC data products. This is the motivation for creating an additional "independent" ocean mass reconstruction (OMrecon) to determine whether GRACE/GRACE-FO models capture the full extent of ocean mass increase post-2016.

By correcting Jason-3 altimetry for an adjusted wet troposphere correction (WTC) and argo for salinity dri^os ocean mass misclosure is reduced although still remains for some ocean basins and at interannual scales which is also consistent with recent work (e.g. Barnoud et al. 2023). It is not obvious to me what about these findings are particularly new or novel. Previous attempts to close the ocean mass budget have achieved closure to within a similar degree and previous work has already highlighted that misclosure is prominent regionally and at interannual timescales. I can however, appreciate the work that has gone into producing the OMrecon, which provides a mostly-independent dataset to compare to GRACE/GRACE-FO. For the most part, their reconstruction matches well the GRACE/GRACEFO observed ocean mass change, supporting their conclusion that GRACE/GRACE-FO is not the source of misclosure.

Summary

The summary needs a rewrite, there are quite a few typos and confusing sentences. I think it is important to highlight that closure of the global ocean mass budget has been a problem but near-closure has already been achieved by accounting for argo biases and WTC from alternative MWR (e.g. Barnoud et al., 2023). However, misclosure of the global ocean mass budget remains at interannual scales and in individual ocean basins.

The summary has been entirely rewritten.

Main

1. OMrecon is referred to as an independent dataset of ocean mass change throughout the paper. I take issue with this because GRACE/GRACE-FO data are used to generate OMrecon in multiple ways by (1) using GRACE-FO interannual variability to extend watergap, (2) using IMBIE combined assessment for Antarctic mass change, and (3) extending the mass balance of peripheral glaciers.

While OMrecon is not entirely independent from GRACE, it's important to note that its reliance on GRACE-FO interannual variability is limited to extending data for glaciers and the Antarctic mass balance.

As of 2020, no publicly available glacier-specific or gridded glacial estimates with global coverage are accessible. As outlined in the methodology section, our global glacier mass balance figures are consistent with those from WGMS. Furthermore, considering the

minimal inter-annual variation in global glacier mass balance (as indicated in Figure 2), we have confidence that any GRACE dependency has had an insignificant impact on the ocean mass contribution results.

Regarding the Antarctic Ice Sheet, the challenge lies in the absence of sufficient estimates that are entirely independent of gravimetry. As a result, we have based our analysis on the best available estimate from IMBIE, acknowledging that GRACE has played a role in deriving this multi-method estimate. The inherent uncertainty is derived from the complete range of Antarctic mass balance estimates, thereby covering the full extent of possible AIS mass balance estimates from other methods. It's worth emphasizing that gravimetry-based mass balance estimates exhibit notable consistency with altimetry-based estimates from 2002 to 2019, as depicted in Figure 3a of Otosaka (2023). The AIS makes shift from mass loss to mass gain in in Figure 1 and 2 around 2020, which has been confirmed by reanalysis models (see comment to reviewer #2), even though the interannual change is still comparably small particularly when compared to natural land water storage (LWS). As a result, we are confident that the GRACE dependency has had minimal impact on our main findings.

2. Paragraph starting line 59 needs a rewrite, split the explanation of how ocean mass changes can be estimated from altimetry and ocean mass reconstructions into separate sentences. Give examples of where these land contributions come from (e.g. land surface models?). My first thought reading this was that these land contributions come from GRACE/GRACE-FO and therefore, this does not provide an “independent estimate”.

In response to the comment from reviewer #2, we have restructured this section to provide a more comprehensive introduction to OMrecon and steric-corrected altimetry.

3. Paragraph starting line 97: you reduce the disclosure of the ocean mass budget by correcting altimetry for the MWR WTC and correcting argo for an “annual global halosteric contribution”. Barnoud et al., 2023 also replaced the argo-based thermosteric component, would this account for the remaining gap in altimetrysteric and GRACE/GFO?

Our findings align with those of Barnoud et al., 2023. Replacing the thermosteric component with the oceanographic models utilized in their study could potentially narrow the gap between steric-corrected altimetry and GRACE-GFO. However, it's essential to note that ORAS5, like ECCO, also incorporates GRACE observations, making it non-independent. Upon examining Figure S4 in Barnoud et al, 2023, a significant portion of the difference between ARGO observations and ORAS5 is attributed to deep steric warming, which undergoes an acceleration from 0.02-0.05 mm/year to over 0.5 mm/year from 2018 to 2020. This rapid increase in global deep steric warming is probably incorrect, as it would otherwise be very concerning. Consequently, we have opted to retain the conventional approach, utilizing unconstrained ARGO observations corrected for the recognized salinity drift. It's worth noting that, as mentioned, a thermosteric drift or bias remains a plausible explanation for some of the regional and global differences observed.

4. Figure 1: I find this figure really difficult to understand, some colours are used multiple times for different data. I also think it would be appropriate to interpolate

between missing GRACE months (leaving the mission gap empty). Tick marks on the left side of the subplots would also help

We understand your concerns about the figure's clarity. As this was also mentioned by reviewer #2, we have edited the figure to enhance its readability.

5. Paragraph starting 169: add numbers to support the statement that "The contrasting pattern can be attributed to the tendencies of the climate reanalysis data...In particular for Africa". Add Australia to ext Fig 1. If the climate reanalysis data underestimates LWS during la nina this will be very obvious compared to the GRACEFO data

We have revised the paragraph and incorporated a figure illustrating the difference between the model and GRACE data in Africa. Additionally, Australia/New Zealand has been separated from Southeast Asia and presented as a standalone subplot in suppl. figure 2 (previous ext. Fig 1), where the interannual difference between GRACE and modelled LWS is visible.

6. Line 190-192: Why consider the difference in OMrecon with/without global mean atmospheric mass?

It was to acknowledge, that atmospheric mass needs accounted for (and is non-negligible) when creating ocean mass budgets for land contributions that are comparable with GRACE. We have removed this sentence now.

7. Line 203: Looking at Fig 2, I'm not convinced of this relationship, the largest deviations between OMrecon and GRACE/altimetry seem to occur before large la nina or el nino periods. Could you provide some statistics to back up your statement or add OMrecon – GRACE to Fig 1a so we can line up where the greatest differences occur with respect to ENSO?

OMrecon – GRACE is shown in Figure 1b, where the dots indicate El Nino/ La Nina periods. We think that it is evident from this figure, that the OMrecon underestimates the short-term climate variability. We have however softened the language regarding its association with ENSO, as the standard correlation test did not find a significant correlation (between 0.3 – 0.6, depending on the method).

8. Line 230: "it is worth noting that there are contributing factors that are also relevant to highlight" what are they?

The sentence was leading to the next paragraph. The sentence is now the lead-in of the next paragraph which has been rewritten to highlight that these 'other contributing factors' are observed by looking at the difference between GRACE and steric-corrected altimetry.

9. Line 240-241: I'm confused by this statement

Also covered by the answer to reviewer #1. This sentence has been edited to clarify what has been done with the halosteric sea level and what it means for regional ocean mass.

Conclusions

line 291-292: This is misleading. GRACE-FO accelerometer errors will not cause drifts in the ocean mass estimates. Biases on the accelerometer measurements are accounted for during gravity field inversion.

We have broadened the sentence to say ‘technical or data-processing’.

Minor:

Line 94: which steric corrected altimetry, there are three plotted in Figure 1? And which subplot of Figure 1?

Sentence changed.

Line 102: extra bracket needs to be removed

Fixed

Line 103: you haven’t closed the post-2016 budget yet. How much is the “small divergent ocean mass change still evident in the GFO era”?

It refers to the previous sentence, where the post-2016 difference with GRACE is 0.29 ± 0.44 mm y^{-1} , thereby within the uncertainty (budget closed), but still quite large trend difference.

We have kept the sentence as it is.

Line105: “CDR” referenced before definition

Fixed.

Line110: two uses of “observed”

Fixed.

Line 110: I think you need to specify in the main text where these time series of land-based water mass change are coming from

We have added some additional info on the source of data in the main text.

Line 120: the IMBIE estimates terminate in December 2020

Changed

Line 188: add ref to Fig 1d

Fixed

Line 190: while a -> while

Fixed

Line 301: regionally -> regional

Fixed

Supplementary

1. GACE-Observations

a. What is GAD

b. Isn’t GAD removed just from ocean mascons anywhere? Therefore, you aren’t using two different solutions.

c. Explicitly say whose solution you are using (GSFC mascons RL06?)

d. Specify the GIA model in text (ICE6G_D)

e. I think it would be appropriate to interpolate over missing GRACE months

All of the above has been added/corrected.

REVIEWERS' COMMENTS

Reviewer #2 (Remarks to the Author):

I think that all my questions have been addressed and my doubts have been solved. Therefore, I recommend publication.

Reviewer #3 (Remarks to the Author):

I commend the authors of “Global and regional ocean mass budget closure since 2003” submitted to Nature Communications for taking the time to address my comments and amend the manuscript. The revised manuscript is a significantly improved version of the original, however, there are some minor comments I would like to see addressed before its publication.

3. Paragraph starting line 97: you reduce the misclosure of the ocean mass budget by correcting altimetry for the MWR WTC and correcting argo for an “annual global halosteric contribution”. Barnoud et al., 2023 also replaced the argo-based thermosteric component, would this account for the remaining gap in alnmetry-steric and GRACE/GFO?

Our findings align with those of Barnoud et al., 2023. Replacing the thermosteric component with the oceanographic models utilized in their study could potentially narrow the gap between steric-corrected altimetry and GRACE-GFO. However, it's essential to note that ORAS5, like ECCO, also incorporates GRACE observations, making it non-independent. Upon examining Figure S4 in Barnoud et al, 2023, a significant portion of the difference between ARGO observations and ORAS5 is attributed to deep steric warming, which undergoes an acceleration from 0.02-0.05 mm/year to over 0.5 mm/year from 2018 to 2020. This rapid increase in global deep steric warming is probably incorrect, as it would otherwise be very concerning. Consequently, we have opted to retain the conventional approach, utilizing unconstrained ARGO observations corrected for the recognized salinity drift. It's worth noting that, as mentioned, a thermosteric drift or bias remains a plausible explanation for some of the regional and global differences observed.

I think this information re deep steric warming is relevant and should be added in some form to the main text.

Line 103: you haven't closed the post-2016 budget yet. How much is the “small divergent ocean mass change still evident in the GFO era”?

It refers to the previous sentence, where the post-2016 difference with GRACE is 0.29 ± 0.44 mm y⁻¹, thereby within the uncertainty (budget closed), but still quite large trend difference. We have kept the sentence as it is.

I still take issue with the wording here as you state you have achieved closure but then immediately follow with the caveat that there is a divergence in the mass change in the GFO era. I suggest rewording to say that that the misclosure is significantly reduced as there is still an obvious divergence post-2016. Alternatively, you could specify for what period GRACE falls within the error bars of altimetry (MWR/CDR) – steric (looks like up to 2020 – hard to tell from Fig 1b)

4. Figure 1: I find this figure really difficult to understand, some colours are used multiple times for different data. I also think it would be appropriate to interpolate between missing GRACE months (leaving the mission gap empty). Tick marks on the other side of the subplots would also help. We understand your concerns about the figure's clarity. As this was also mentioned by reviewer #2, we have edited the figure to enhance its readability.

Figure 1 is significantly improved. It could be improved further if the different error envelopes for each of the altimetry-derived ocean mass curves could be distinguished from each other in b.

7. Line 203: Looking at Fig 2, I'm not convinced of this relationship, the largest deviations between OMrecon and GRACE/altimetry seem to occur before large la nina or el nino periods. Could you provide some statistics to back up your statement or add OMrecon – GRACE to Fig 1a so we can line up where the greatest differences occur with respect to ENSO?

OMrecon – GRACE is shown in Figure 1b, where the dots indicate El Nino/ La Nina periods. We think that it is evident from this figure, that the OMrecon underestimates the short-term climate variability. We have however softened the language regarding its association with ENSO, as the standard correlation test did not find a significant correlation (between 0.3 – 0.6, depending on the method).

The wording is now appropriate. Fig 2 would be improved by the addition of a grey bar showing GRACE-GFO gap to b.

Minor

Line 120-122 suggested addition to improve readability

“Mass-balance time series of glaciers and the Antarctic Ice Sheet terminates in December 2018 and 2020 respectively, and thus a slightly shorter timespan is available compared to the GRACE observational record.”

Line 241-249: GRACE is 0.45 mm/yr lower in the North Atlantic (not 1.45 mm/yr). Further, the difference in the south Atlantic is not the “opposite”, here, grace estimates a greater ocean mass by 1.33 mm/yr compared to altimetry-steric which is almost triple the difference in the North Atlantic. Simply combining the two regions won't close the budget here as stated in the paragraph starting Line 248.

Fig 3: perhaps adding the full Atlantic integration would support the claim in Line 248 that adding the North and South would close the regional ocean mass budget.

Line 236: “near linear pattern” > “near linear mass increase”

Line 237: “show greater variation” > “show greater interannual variation”

Line 258: wording: “that opposite the dynamic...” -> “that oppose the dynamic..”

Line 277: temporally -> temporarily ?

Supplementary: Reorder your supplementary figures so they are mentioned chronologically in the main text, they are currently mentioned in the order SF1, SF4, SF3, SF2, SF5-7

Response to reviewer comments

Old comments by author in red

New comments by author in dark red

Reviewer 2:

I think that all my questions have been addressed and my doubts have been solved. Therefore, I recommend publication.

We thank the reviewer for the helpful comments made in the previous review.

Reviewer 3:

I commend the authors of “Global and regional ocean mass budget closure since 2003” submitted to Nature Communications for taking the time to address my comments and amend the manuscript. The revised manuscript is a significantly improved version of the original, however, there are some minor comments I would like to see addressed before its publication.

We appreciate the reviewer's insightful feedback provided in the previous review. We hope that our last revisions have effectively addressed the comments in this final review.

3. Paragraph starting line 97: you reduce the misclosure of the ocean mass budget by correcting altimetry for the MWR WTC and correcting argo for an “annual global halosteric contribution”. Barnoud et al., 2023 also replaced the argo-based thermosteric component, would this account for the remaining gap in alnmetry-steric and GRACE/GFO?

Our findings align with those of Barnoud et al., 2023. Replacing the thermosteric component with the oceanographic models utilized in their study could potentially narrow the gap between steric-corrected altimetry and GRACE-GFO. However, it's essential to note that ORAS5, like ECCO, also incorporates GRACE observations, making it non-independent. Upon examining Figure S4 in Barnoud et al, 2023, a significant portion of the difference between ARGO observations and ORAS5 is attributed to deep steric warming, which undergoes an acceleration from 0.02-0.05 mm/year to over 0.5 mm/year from 2018 to 2020. This rapid increase in global deep steric warming is probably incorrect, as it would otherwise be very concerning. Consequently, we have opted to retain the conventional approach, utilizing unconstrained ARGO observations corrected for the recognized salinity drift. It's worth noting that, as mentioned, a thermosteric drift or bias remains a plausible explanation for some of the regional and global differences observed.

I think this information re deep steric warming is relevant and should be added in some form to the main text.

We have revised the main text to clarify that the steric signal is computed down to ocean depths of 5400 metres. In the same revision, we cite the work of Chang et al (2019) to note that below 2000 m, the deep ocean is showing near constant warming which has a non-

negligible contribution to steric sea level change, and cross reference a figure in the supplementary material to support this.

Line 103: you haven't closed the post-2016 budget yet. How much is the "small divergent ocean mass change still evident in the GFO era"?

It refers to the previous sentence, where the post-2016 difference with GRACE is 0.29 ± 0.44 mm y^{-1} , thereby within the uncertainty (budget closed), but still quite large trend difference. We have kept the sentence as it is.

I still take issue with the wording here as you state you have achieved closure but then immediately follow with the caveat that there is a divergence in the mass change in the GFO era. I suggest rewording to say that that the misclosure is significantly reduced as there is still an obvious divergence post-2016. Alternatively, you could specify for what period GRACE falls within the error bars of altimetry (MWR/CDR) – steric (looks like up to 2020 – hard to tell from Fig 1b)

We agree with the reviewer and have changed the wording accordingly.

4. Figure 1: I find this figure really difficult to understand, some colours are used multiple times for different data. I also think it would be appropriate to interpolate between missing GRACE months (leaving the mission gap empty). Tick marks on the other side of the subplots would also help

We understand your concerns about the figure's clarity. As this was also mentioned by reviewer #2, we have edited the figure to enhance its readability.

Figure 1 is significantly improved. It could be improved further if the different error envelopes for each of the altimetry-derived ocean mass curves could be distinguished from each other in b.

We have changed the colors (one with lighter (halosteric uncorrected) color and one with a darker color (MWR) than the original color of MWR/CDR), which makes it easier to distinguish the shaded areas.

7. Line 203: Looking at Fig 2, I'm not convinced of this relationship, the largest deviations between OMrecon and GRACE/altimetry seem to occur before large la nina or el nino periods. Could you provide some statistics to back up your statement or add OMrecon – GRACE to Fig 1a so we can line up where the greatest differences occur with respect to ENSO?

OMrecon – GRACE is shown in Figure 1b, where the dots indicate El Nino/ La Nina periods. We think that it is evident from this figure, that the OMrecon underestimates the short-term climate variability. We have however softened the language regarding its association with ENSO, as the standard correlation test did not find a significant correlation (between 0.3 – 0.6, depending on the method).

The wording is now appropriate. Fig 2 would be improved by the addition of a grey bar showing GRACE-GFO gap to b.

Figure 2 has been changed with the addition of a grey bar showing the GRACE-GFO gap.

Minor

Line 120-122 suggested addition to improve readability

“Mass-balance time series of glaciers and the Antarctic Ice Sheet terminates in December 2018 and 2020 respectively, and thus a slightly shorter timespan is available compared to the GRACE observational record.”

Corrected

Line 241-249: GRACE is 0.45 mm/yr lower in the North Atlantic (not 1.45 mm/yr). Further, the difference in the south Atlantic is not the “opposite”, here, grace estimates a greater ocean mass by 1.33 mm/yr compared to altimetry-steric which is almost triple the difference in the North Atlantic. Simply combining the two regions won’t close the budget here as stated in the paragraph starting Line 248.

We respectfully disagree with the comment. GRACE data indicates a lower ocean mass trend in the North Atlantic (1.40 mm/yr) compared to the South Atlantic (2.85 mm/yr), equal to a 1.45 mm/yr difference, which is opposite to the pattern observed in steric-corrected altimetry (higher in the North Atlantic at 1.85 mm/yr than in the South Atlantic at 1.52 mm/yr). Also, the difference between GRACE and steric-corrected altimetry is negative in the North Atlantic and positive in the South Atlantic. Referring to Figure 3 and assuming equal weight for both the North and South Atlantic, the combined ocean mass trend estimates for the Atlantic are: 1.96 – 2.28 mm/yr (GRACE), 1.88 – 2.26 mm/yr (OMrecon), and 1.31 – 2.06 mm/yr (steric-corrected altimetry). These estimates overlap, and the budget is balanced within the uncertainty range (1 standard deviation) for all three methods, and not solely OMrecon and GRACE as initially mentioned. Therefore, we have updated our statement to indicate that the combined Atlantic analysis reconciles the budget across all estimates.

Fig 3: perhaps adding the full Atlantic integration would support the claim in Line 248 that adding the North and South would close the regional ocean mass budget.

We have incorporated the combined Atlantic numbers into the text. We believe that adding a full Atlantic panel to Figure 3 would just duplicate information that is already conveyed in the existing Figure hence have chosen to maintain the Figure as is.

Line 236: “near linear pattern” > “near linear mass increase”

Line 237: “show greater variation” > “show greater interannual variation”

Line 258: wording: “that opposite the dynamic...” -> “that oppose the dynamic..”

Line 277: temporally -> temporarily?

The above suggestions for rewording have been implemented.

Supplementary: Reorder your supplementary figures so they are mentioned chronologically in the main text, they are currently mentioned in the order SF1, SF4, SF3, SF2, SF5-7

The supplementary figures have been reordered.